# p53 ensures the normal behavior and modification of G1/S-specific histone H3.1 in the nucleus

Tsukasa Oikawa[1] , Junya Hasegawa[2] , Haruka Handa[1], Naomi Ohnishi[3], Yasuhito Onodera[1,4], Ari Hashimoto[1] , Junko Sasaki[2], Takehiko Sasaki[2], Koji Ueda[3], Hisataka Sabe[1,5]

H3.1 histone is predominantly synthesized and enters the nucleus during the G1/S phase of the cell cycle, as a new component of duplicating nucleosomes. Here, we found that p53 is necessary to secure the normal behavior and modification of H3.1 in the nucleus during the G1/S phase, in which p53 increases C-terminal domain nuclear envelope phosphatase 1 (CTDNEP1) levels and decreases enhancer of zeste homolog 2 (EZH2) levels in the H3.1 interactome. In the absence of p53, H3.1 molecules tended to be tethered at or near the nuclear envelope (NE), where they were predominantly trimethylated at lysine 27 (H3K27me3) by EZH2, without forming nucleosomes. This accumulation was likely caused by the high affinity of H3.1 toward phosphatidic acid (PA). p53 reduced nuclear PA levels by increasing levels of CTDNEP1, which activates lipin to convert PA into diacylglycerol. We moreover found that the cytosolic H3 chaperone HSC70 attenuates the H3.1-PA interaction, and our molecular imaging analyses suggested that H3.1 may be anchored around the NE after their nuclear entry. Our results expand our knowledge of p53 function in regulation of the nuclear behavior of H3.1 during the G1/S phase, in which p53 may primarily target nuclear PA and EZH2.

## Introduction

Molecular processes involved in the nuclear transport and intranuclear behavior of histones, and those involved in the engagement of histones with the nucleosome and their epigenetic marking are thought to be closely associated with genome regulation and integrity. From this perspective, molecular chaperones and transporters involved in the nuclear transport of H3 histones have been extensively studied (Tyler et al, 1999; Mühlhäusser et al, 2001; Groth et al, 2005; Campos et al, 2010; Alvarez et al, 2011; Campos et al, 2015; Soniat et al, 2016; Apta-Smith et al, 2018; Pardal & Bowman, 2022). H3.1 histone is

predominantly synthesized and enters the nucleus during the G1/S phase of the cell cycle (Mendiratta et al, 2019). Histone chaperone networks, including HSC70 and chromatin assembly factor 1 (CAF1), have been shown to guide H3.1 from the cytosol to sites of DNA replication (Tagami et al, 2004; Campos et al, 2010; Pardal & Bowman, 2022). However, there are still many missing links in our understanding of the nuclear regulation and behavior of histones.

To understand the nuclear regulation and behavior of histones, we here focused on H3.1, because of its specificity to the G1/S phase of the cell cycle, which may be able to simplify experimental settings compared with the analysis of other histones. We found that p53, a product of the tumor suppressor gene *TP53*, is integral for ensuring the normal behavior of H3.1 in the nucleus, whereas p53 does not appear to be necessary for the nuclear entry of H3.1. Our results demonstrate a novel association between p53 and H3.1, in which p53 is required for H3.1 not to be trapped around the nuclear envelope (NE), and not to be therein marked by enhancer of zeste homolog 2 (EZH2) as suppressive, without forming nucleosomes. For this function, p53 appears to down-regulate nuclear phosphatidic acid (PA) levels, likely via its transcriptional activity, and exclude EZH2 from the H3.1 interactome.

## Results

### Loss of p53 causes the perinuclear accumulation of lysine 27-trimethylated histone H3 (H3K27me3)

Using super-resolution three-dimensional structured illumination microscopy (SIM), we found that silencing of *TP53* by its siRNAs (si*TP53*) in normal human mammary epithelial HMLE cells causes the accumulation of H3K27me3, but not lysine 4-trimethylated histone H3 (H3K4me3) or lysine 27-acetylated histone H3 (H3K27ac) near the NE (Figs 1A–C and S1A). H3K27me3, H3K4me3, and H3K27ac

---

[1]Department of Molecular Biology, Graduate School of Medicine, Hokkaido University, Sapporo, Japan    [2]Department of Biochemical Pathophysiology/Lipid Biology, Medical Research Institute, Tokyo Medical and Dental University, Bunkyo-ku, Japan    [3]Cancer Proteomics Group, Cancer Precision Medicine Center, Japanese Foundation for Cancer Research, Koto-ku, Japan    [4]Global Center for Biomedical Science and Engineering, Graduate School of Medicine, Hokkaido University, Sapporo, Japan    [5]Institute for Genetic Medicine, Hokkaido University, Sapporo, Japan

Correspondence: oikawa_tsukasa@med.hokudai.ac.jp; sabeh@med.hokudai.ac.jp

---

 

were spread throughout the nucleus in control cells (i.e., si*Scr*-treated cells) (Fig 1A). si*TP53* treatment of the human cancer cell lines A549 and MCF7 also caused similar perinuclear accumulation of H3K27me3 (Figs 1C and S1A and B). Levels of these histone modifications were not notably affected by si*TP53* (Fig 1D). Similar accumulation of H3K27me3 was also observed in the human cancer cell line H1299, which lacks p53 expression, and in the mouse embryonic fibroblast cell line MB352 established from *Trp53*$^{-/-}$ mice (Figs 1E and S1A). We confirmed that the introduction of wild-type p53 (p53 WT) in MB352 cells cancels the H3K27me3 accumulation (Fig S1A, C, and D). Therefore, p53 deficiency may cause the aberrant perinuclear accumulation of H3K27me3 in different types of cells, including cancer cells, in humans and mice.

### Perinuclear accumulation of H3K27me3 occurs during the G1/S phase

We then investigated whether H3K27me3 accumulation occurs during a specific phase of the cell cycle. H1299 cells and MB352 cells stably expressing the cell-cycle indicator Fucci (Sakaue-Sawano et al, 2008) demonstrated that H3K27me3 accumulation occurs predominantly from the late G1 to the early S phase in the absence of p53 (Figs 1F and S1A). Thymidine blockade of G1/S progression in H1299 and MB352 cells also caused H3K27me3 accumulation (Figs 1G and S1A and E). Therefore, the perinuclear accumulation of H3K27me3 in the absence of p53 appeared to occur predominantly during the G1/S phase.

### p53 suppresses H3K27me3 accumulation without cell-cycle arrest or apoptosis

The expression of p53 WT significantly decreased H3K27me3 accumulation in thymidine-blocked H1299 cells and MB352 cells (Figs 1G and S1A and D). However, p53 is known to induce genes involved in cell-cycle arrest or apoptosis (el-Deiry et al, 1993) (Figs 1H and S1E). We then used a p53 mutant, p53 QM, bearing the L22Q/W23S/W53Q/F54S mutations, which was previously shown to not induce the expression of such genes (see Fig S1D) (Venot et al, 1999). We here confirmed that p53 QM did not induce genes known to be involved in cell-cycle arrest, apoptosis, and metabolism, when expressed in H1299 cells and MB352 cells (Figs 1H, S1D and E, and S2, and Table S1). Nevertheless, p53 QM substantially suppressed the accumulation of H3K27me3 in these cells (Figs 1G and I and S1A), whereas it did not affect the localization of another NE-associated histone, lysine 9-trimethylated histone H3 (H3K9me3) (Fig S1F). Furthermore, the proximity ligation assay (PLA) coupled with SIM imaging in thymidine-blocked H1299 cells demonstrated that the expression of p53 QM significantly reduces the colocalization of H3K27me3 with lamin A/C near the NE (Fig 1J and K). Therefore, p53 appears to suppress the aberrant perinuclear accumulation of H3K27me3 and its colocalization with lamin A/C during the G1/S phase, likely independent of cell-cycle arrest, apoptosis, and metabolic remodeling.

### Properties of perinuclearly accumulated H3K27me3 histones

We then investigated the properties of the perinuclearly accumulated H3K27me3 histones. Costaining of H3K27me3 with lamin A/C in H1299 cells showed their close colocalization with the NE (Fig 2A). Their costaining with DNA suggested that these histones do not colocalize well with DNA (Fig 2B). We then visualized the DNA replication sites using biotin-labeled deoxyuridine triphosphate (dUTP) (Maya-Mendoza et al, 2012; Alabert et al, 2014). SIM imaging of H1299 cells, which were blocked once with thymidine and then released for 20 min, revealed that most of the perinuclearly accumulated H3K27me3 histones were not localized to DNA replication sites, whereas the localization of H3K27me3 histones to DNA replication sites was clearly detected in the presence of p53 QM (Fig 2C and D).

We furthermore analyzed the possible nucleosome formation of these histones. PLA between H3K27me3 and H4 in thymidine-blocked H1299 cells demonstrated that most of the PLA signals were found inside the nucleoplasm, rather than near the NE (Fig 2E and F). Furthermore, the levels of these signals near the NE were not notably changed, irrespective of p53 QM (Fig 2E and F). Thus, most of the H3K27me3 histones accumulated near the NE in the absence of p53 did not appear to have formed nucleosomes, bound to DNA, or localized to DNA replication sites.

### H3.1/H3.2 comprises the accumulated H3K27me3

H3.1 and H3.2 are synthesized in the cytosol, and then transported into the nucleus during the G1/S phase of the cell cycle, whereas H3.3 acts independently of DNA replication (Tagami et al, 2004; Mendiratta et al, 2019). These H3 variants were expressed at similar levels in H1299 cells, irrespective of p53 QM expression or thymidine blockade (Fig 3A). Using antibodies specific to H3.1/H3.2 and to H3.3, we then found that H3.1/H3.2, but not H3.3, accumulated near the NE in thymidine-blocked H1299 cells (Fig 3B) and that such accumulation was largely mitigated by p53 QM (Fig 3C). PLA coupled with SIM imaging in thymidine-blocked H1299 cells demonstrated that the expression of p53 QM significantly reduces the colocalization of H3.1/H3.2 with lamin A/C near the NE (Fig 3D and E). Thus, H3.1/H3.2, but not H3.3, appear to predominantly comprise the perinuclearly accumulated H3K27me3 histones. Antibodies that selectively recognize either H3.1 or H3.2 were not available, and it hence remains unclear as to which of them comprise the accumulated H3K27me3 histones.

### p53 increases CTDNEP1 and decreases EZH2 levels in the nuclear H3.1 interactome

We then sought to obtain a clue to link p53 with H3.1/H3.2. For this purpose, we performed an interactome analysis using liquid chromatography–mass spectrometry (LC-MS) on anti-H3 immunoprecipitants from the nuclear fraction of thymidine-blocked H1299 cells (Fig 3F). Proteins were crosslinked with dithiobis (succinimidyl propionate) (DSP) before solubilization. Notably, we found that the anti-H3 antibody predominantly pulled down H3.1 under this condition, rather than H3.2 or H3.3 (Fig 3G and Table S2), although we do not have a clear explanation for this. Among the different proteins that were coprecipitated, the amount of

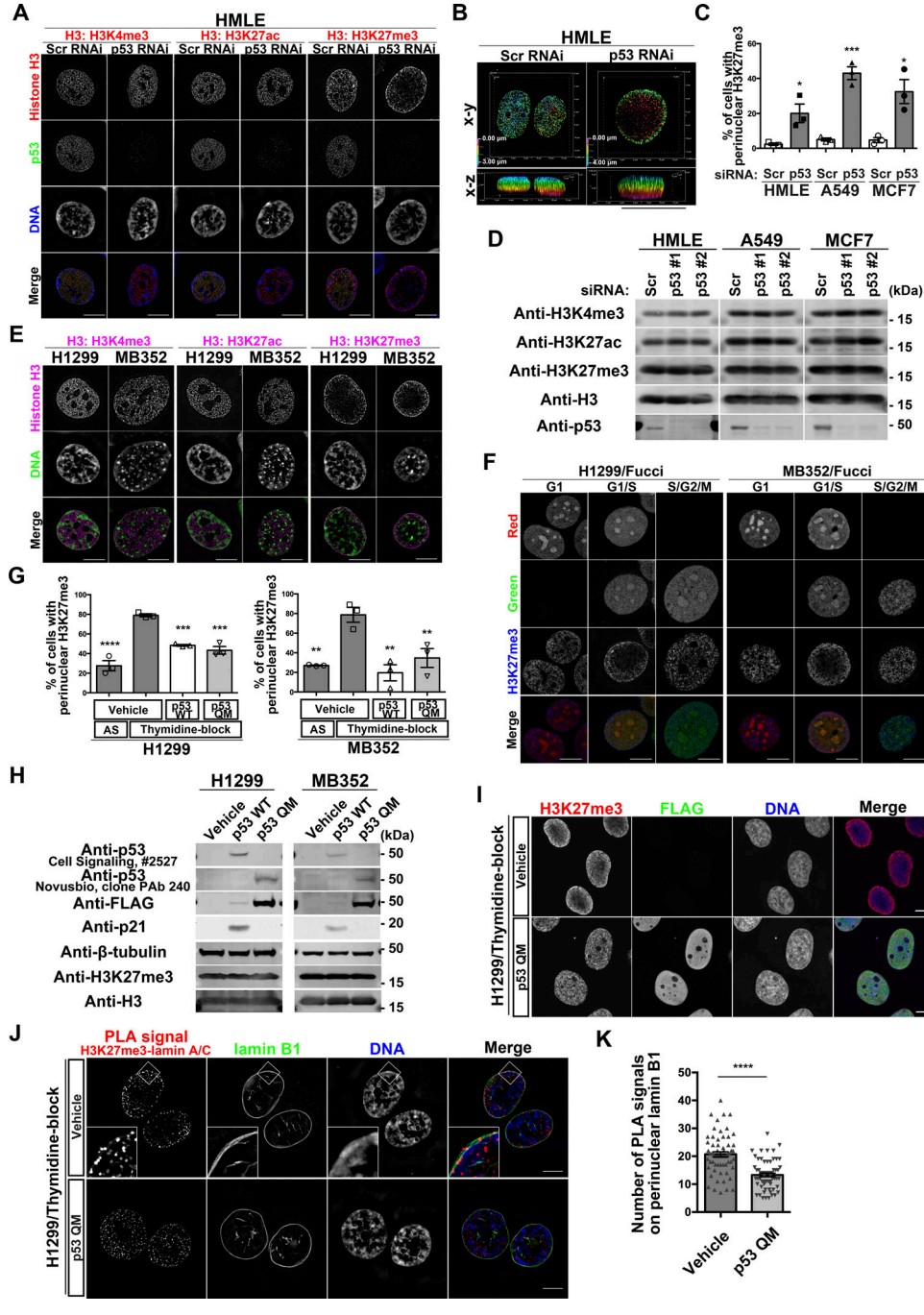

**Figure 1. p53 suppresses perinuclear H3K27me3 during the G1/S phase.**
**(A)** Representative structured illumination microscopy (SIM) images of HMLE cells showing the localization of H3K4me3, H3K27ac, or H3K27me3 (red), p53 (green), and DNA (blue). Individual fluorescence channels are shown in grayscale with merged images on the bottom. Bars, 10 $\mu$m. **(B)** Depth-coded alpha-blending views of H3K27me3 from the 3D-SIM images of HMLE cells. Bar, 10 $\mu$m. **(C)** Quantification of perinuclear H3K27me3 accumulation in the indicated cells. n = 3 biological replicates, ≥100 cells per replicate; ***$P$ = 0.0006 and *$P$ < 0.03, two-tailed unpaired $t$ test. **(D)** Representative immunoblots of the indicated antibodies. **(E)** Representative SIM images showing the localization of H3K4me3, H3K27ac, or H3K27me3 (magenta) and DNA (green) in the indicated cells. Individual fluorescence channels are shown in grayscale with merged images on the bottom. Bars, 10 $\mu$m. **(F)** Representative SIM images showing the localization of H3K27me3 (blue) in different stages of the cell cycle, in H1299 cells and MB352 cells. The Fucci fluorescent probe labels individual G1-phase nuclei in red and S/G2/M-phase nuclei in green. Individual fluorescence channels are shown in grayscale with merged images on the bottom. Bars, 10 $\mu$m. **(G)** Quantification of the indicated cells with perinuclear H3K27me3 accumulation. AS, asynchronous. n = 3 biological replicates, ≥50 cells per replicate; ****$P$ < 0.0001, ***$P$ < 0.001, and **$P$ < 0.01, one-way ANOVA followed by Dunnett's multiple comparisons test. **(H)** Representative immunoblots of the indicated antibodies. **(I)** Representative confocal images of H1299 cells showing the localization of H3K27me3 (red), FLAG (green), and DNA (blue). Individual fluorescence channels are shown in grayscale with merged images on the right. Bars, 10 $\mu$m. **(J)** Representative SIM images of H1299 cells showing the localization of proximity ligation assay signals between H3K27me3 and lamin A/C (red), lamin B1 (green), and DNA (blue). Individual fluorescence channels are shown in grayscale with merged images on the right. Bars, 10 $\mu$m. **(K)** Quantification of the proximity ligation assay spots that overlap with perinuclear lamin B1 in each nucleus. n = 3 biological replicates, ≥60 nuclei; ****$P$ < 0.0001, two-tailed unpaired $t$ test. Source data are available for this figure.

C-terminal domain nuclear envelope phosphatase 1 (CTDNEP1) was more than fourfold higher in the presence of p53 QM than in its absence (Fig 3G and Table S2). EZH2 was also coprecipitated, but its amount was about twofold lower in the presence of p53 than in its absence (Fig 3G and Table S2). The amount of H3.1 in the immunoprecipitants was not notably changed irrespective of the presence/absence of p53 QM, and p53, when present, was coprecipitated with H3.1 (Fig 3G and Table S2). Thus, p53 may play a role in increasing CTDNEP1 and decreasing EZH2 levels in the nuclear H3.1 interactome formed during the G1/S phase.

## p53 suppresses the perinuclear interaction of H3.1 with EZH2

We then focused on EZH2. EZH2 is the catalytic subunit of polycomb repressive complex 2, which generates H3K27me3 (Cao et al, 2002). We confirmed the above results by Western blotting, in which a larger amount of EZH2 was detected in anti-H3 immunoprecipitants from the G1/S-arrested cells in the absence of p53 QM (Fig 4A). Benzonase is a potent endonuclease, and Benzonase treatment during immunoprecipitation did not affect the amount of coprecipitated EZH2, avoiding the possible contamination of nuclear DNA (Fig 4A).

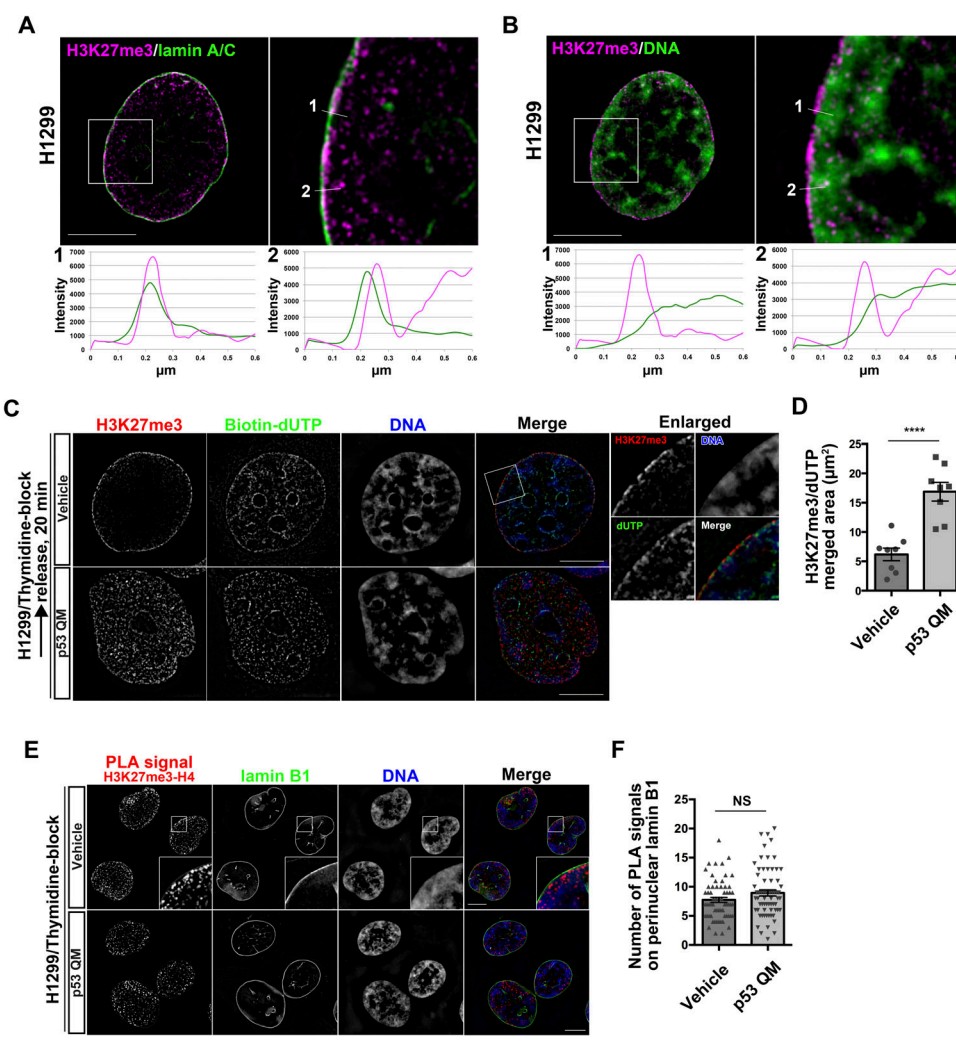

**Figure 2. Perinuclear H3K27me3 does not form nucleosomes.**

**(A, B)** Representative structured illumination microscopy (SIM) images of a H1299 cell showing the localization of H3K27me3 (magenta) and lamin A/C (green) (A), or H3K27me3 (magenta) and DNA (green) (B). Enlarged images of the white boxed areas are shown on the right. Pixel intensities on the traversing white lines (1 and 2) are shown at the bottom. Bars, 10 μm.

**(C)** Representative SIM images of H1299 cells showing the localization of H3K27me3 (red), biotin–dUTP (green), and DNA (blue). Individual fluorescence channels are shown in grayscale with merged images on the right. Enlarged images of the white boxed area are shown on the right. Bars, 10 μm.

**(D)** Quantification of the area of H3K27me3 and dUTP colocalization. n = 2 biological replicates; ****P < 0.0001, two-tailed unpaired t test. **(E)** Representative SIM images of H1299 cells showing the localization of the proximity ligation assay signals between H3K27me3 and H4 (red), lamin B1 (green), and DNA (blue). Individual fluorescence channels are shown in grayscale with merged images on the right. Bars, 10 μm.

**(F)** Quantification of the proximity ligation assay spots that overlap with perinuclear lamin B1 in each nucleus. n = 3 biological replicates, ≥60 nuclei; NS, not significant, two-tailed unpaired t test.

siEZH2 substantially mitigated the perinuclear accumulation of H3K27me3 in these thymidine-blocked H1299 cells (Fig 4B). Western blot analysis showed that siEZH2 reduced the total cellular amounts of H3K27me3, but not H3K4me3 (Fig 4C). However, a small amount of H3K27me3 was still observed in some cells after siEZH2 treatment (Fig 4B). This may be owing to a delayed decrease in H3K27me3 after EZH2 silencing (Fig 4D). In addition, the PLA showed that p53 QM significantly suppresses the colocalization between EZH2 and H3 near the NE in G1/S-arrested cells (Fig 4E and F). Therefore, taken together, it is likely that EZH2 is responsible for the perinuclear accumulation of H3K27me3, whereas p53 appears to suppress the perinuclear interaction between H3 and EZH2.

### H3.1 molecules are tethered around the NE after their nuclear entry

There are two possibilities for the mechanism underlying the accumulation of such H3K27me3 histones that are mostly composed of H3.1/H3.2 around the NE; that is, either they were trapped by the NE when entering the nucleus during the G1/S phase, or they were tethered around the NE after entering the nucleus. To understand which possibility is more likely, we generated H1299 cells with the gene encoding Dronpa, a fluorescent protein with reversible photobleaching/photoactivation (Ando et al, 2004) at the 3'-end of the HIST1H3A gene (encoding H3.1), using the microhomology-mediated end-joining (MMEJ)–assisted knock-in system (Sakuma et al, 2016) (Fig 4G). We then performed FRAP analysis of H3.1-Dronpa in the nucleus (see the Materials and Methods section). After photobleaching the nucleus of a thymidine-blocked H1299 cell, we observed a gradual increase in H3.1-Dronpa signals both at the nuclear periphery and inside the nucleus in 30–40 min (Fig 4H, Video 1, Video 2, and Video 3). We observed similar recovery dynamics of the H3.1-Dronpa signals after photobleaching in the presence of p53 QM, although the perinuclear accumulation of H3.1-Dronpa was not evident (Fig 4I, Video 4, Video 5, and Video 6). Therefore, it is likely that H3.1 molecules are tethered near the NE after entering the nucleus.

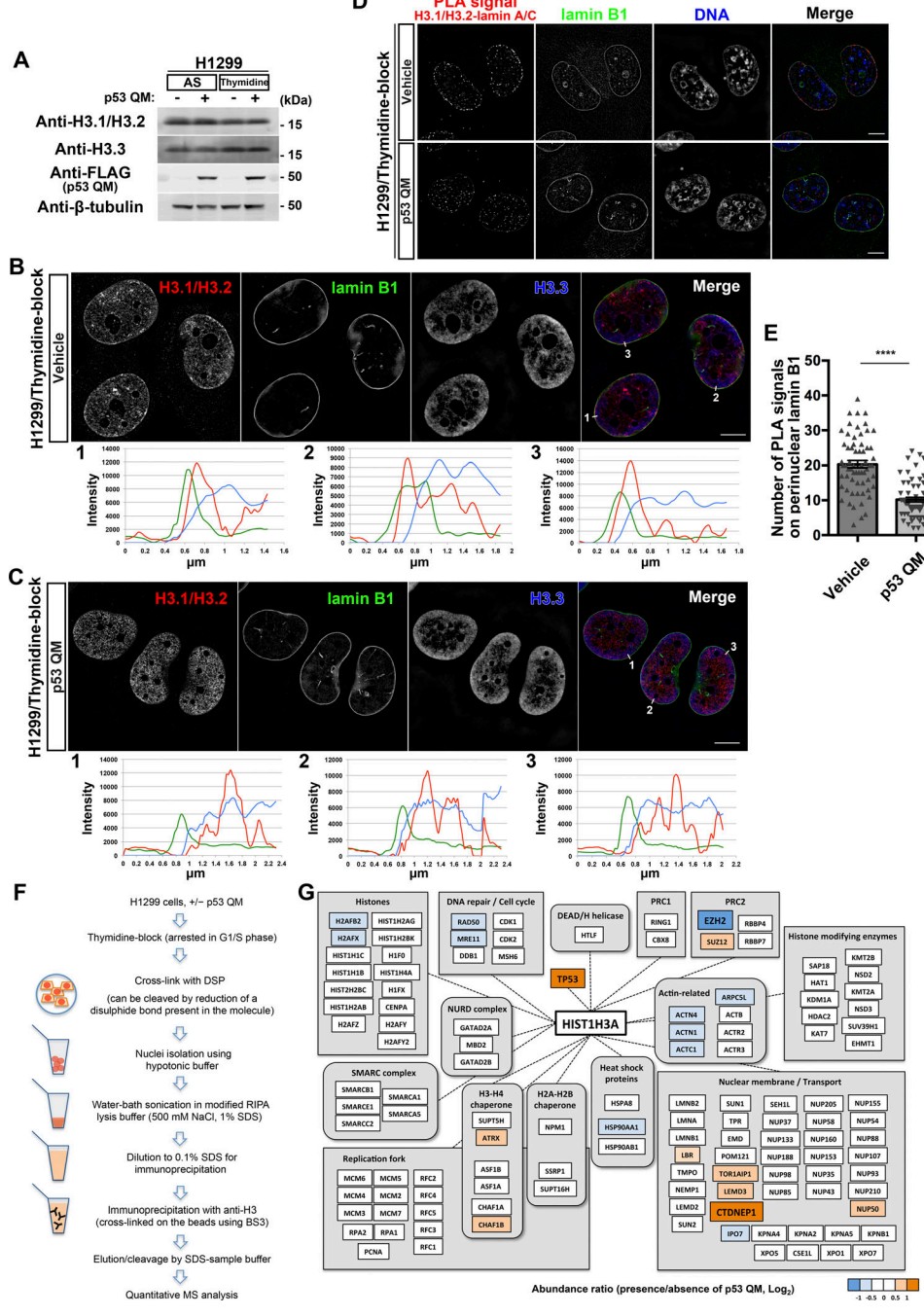

**Figure 3. H3.1/H3.2 comprises perinuclearly accumulated H3K27me3.**
**(A)** Representative immunoblots of the indicated antibodies. AS, asynchronous. **(B, C)** Representative structured illumination microscopy images of H1299 cells showing the localization of H3.1/H3.2 (red), lamin B1 (green), and H3.3 (blue) in the absence (B) or presence (C) of p53 QM. Pixel intensities on the traversing white lines (1, 2, and 3) are shown at the bottom. Individual fluorescence channels are shown in grayscale with merged images on the right. Bars, 10 $\mu$m. **(D)** Representative structured illumination microscopy images of H1299 cells showing the localization of the proximity ligation assay signals between H3.1/H3.2 and lamin A/C (red), lamin B1 (green), and DNA (blue). Individual fluorescence channels are shown in grayscale with merged images on the right. Bars, 10 $\mu$m. **(E)** Quantification of the proximity ligation assay spots that overlap with perinuclear lamin B1 in each nucleus. n = 3 biological replicates, ≥60 nuclei; ****$P <$ 0.0001, two-tailed unpaired $t$ test. **(F)** Schematic representation of H3 immunoprecipitation followed by quantitative MS analysis. **(G)** Representative proteins identified from the MS analysis. The relative abundance of the proteins is presented as a ratio (Log$_2$) in the presence/absence of p53 QM.
Source data are available for this figure.

## p53 uses CTDNEP1 to reduce nuclear PA levels

We next focused on CTDNEP1, which was increased in the H3.1 interactome by p53. CTDNEP1 converts PA to diacylglycerol (DAG) via activation of lipin PA phosphatases (Han et al, 2012; Su et al, 2014; Bahmanyar & Schlieker, 2020; Barger et al, 2022). PA is a minor component of nuclear lipids, and mostly localizes to the NE (Bahmanyar & Schlieker, 2020). We found that H3.1 has a strong affinity to PA, but not to DAG (Fig 5A). Therefore, PA may cause the accumulation of H3.1 around the NE. We then tested whether CTDNEP1 reduces nuclear PA levels and the perinuclear accumulation of H3K27me3. We performed mass spectrometry analysis of nuclear PA levels using nuclear phosphatidylethanolamine (PE) as an internal control, and found that nuclear PA levels (i.e., the PA/PE ratio) in thymidine-blocked H1299 cells are significantly lower in the presence of p53 QM than in its absence (Fig 5B), and that si*CTDNEP1*

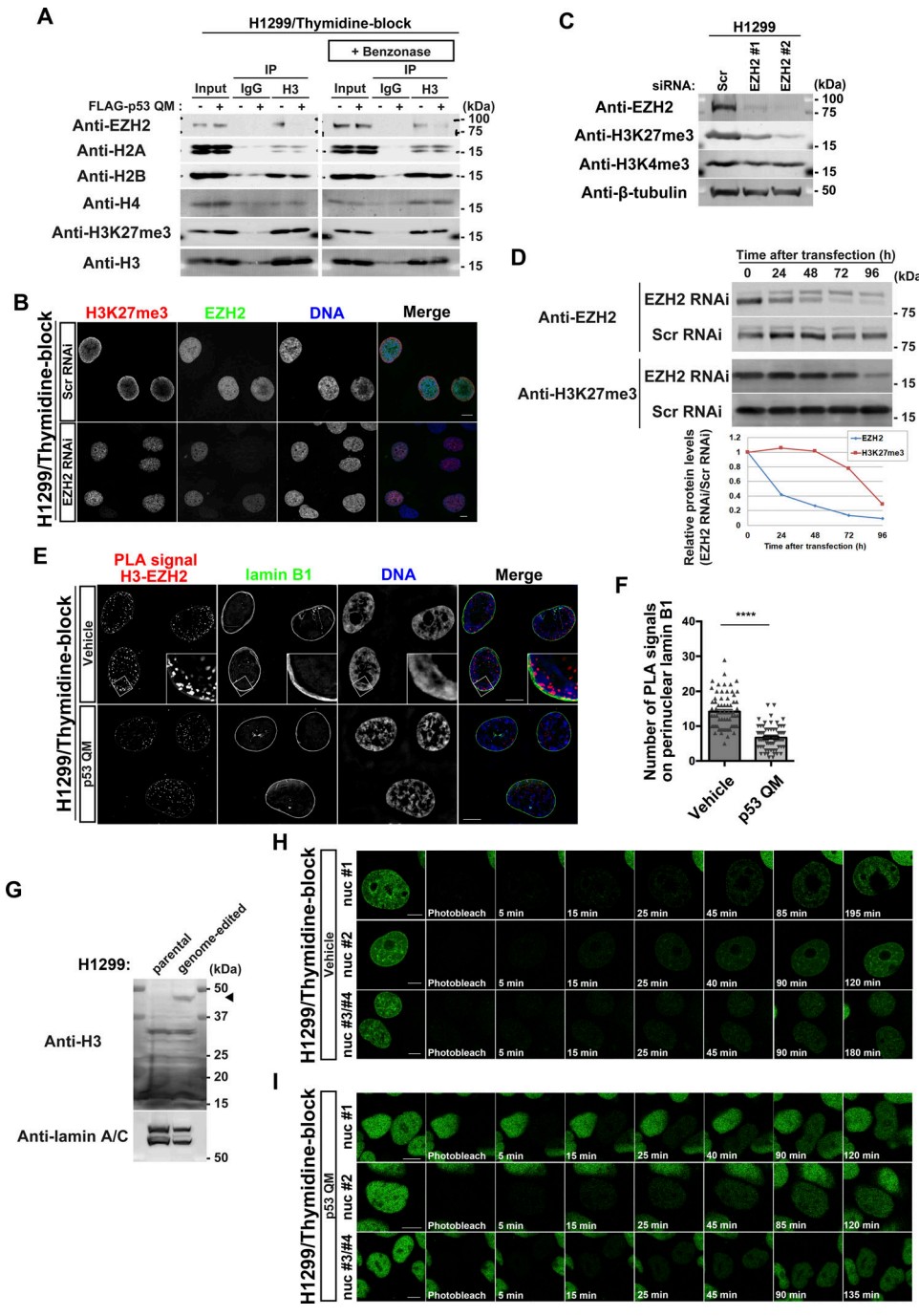

**Figure 4. p53 reduces the proximity between enhancer of zeste homolog 2 (EZH2) and perinuclearly tethered H3.**
**(A)** Representative immunoblots of the indicated antibodies. IP, immunoprecipitation. **(B)** Representative confocal images of H1299 cells showing the localization of H3K27me3 (red), EZH2 (green), and DNA (blue). Cells were fixed at 48 h after siRNA transfection. Individual fluorescence channels are shown in grayscale with merged images on the right. Bars, 10 µm. **(C)** H1299 cells were transfected with the indicated siRNAs and were cultured for 96 h before being subjected to immunoblot analysis. Representative immunoblots of the indicated antibodies. **(D)** H1299 cells were transfected with the indicated siRNAs and were cultured for the indicated hours before being subjected to immunoblot analysis. Representative immunoblots of the indicated antibodies. Normalized relative protein levels (EZH2 RNAi/Scr RNAi) are shown at the bottom. **(E)** Representative structured illumination microscopy images of H1299 cells showing the localization of proximity ligation assay signals between H3 and EZH2 (red), lamin B1 (green), and DNA (blue). Individual fluorescence channels are shown in grayscale with merged images on the right. Bars, 10 µm. **(F)** Quantification of the proximity ligation assay spots that overlap with perinuclear lamin B1 in each nucleus. n = 3 biological replicates, ≥60 nuclei; ****P < 0.0001, two-tailed unpaired t test. **(G)** Representative immunoblots of the indicated antibodies. The arrowhead indicates H3.1-Dronpa. **(H, I)** Representative confocal images showing the localization of H3.1-Dronpa in the absence (H) or presence (I) of p53 QM in H1299 cells. Bars, 10 µm. Source data are available for this figure.

treatment cancels this reduction (Figs 5B and S3A) and restores the perinuclear accumulation of H3K27me3 (Figs 5C and D and S1A). As a control, we confirmed that si*CTDNEP1* treatment in p53-deficient H1299 cells does not notably affect nuclear PA levels (Figs 5E and S3A).

Then, the question arises as to how H3.1 can pass through the NE when p53 is absent and PA levels increase. HSC70 (encoded by *HSPA8*) is a chaperone for H3 (Campos et al, 2010), and was included within the H3.1 interactome irrespective of the presence of p53 QM (see Fig 3G). We found that HSC70 attenuates the interaction of H3.1

with PA (Fig 5F). Therefore, HSC70 may assist H3.1 to smoothly enter the nucleus even in the absence of p53.

The nuclear levels of lipin 1 were also increased upon p53 QM expression in thymidine-blocked H1299 cells, and si*CTDNEP1* canceled this augmentation (Figs 5G and S3A). A biochemical analysis confirmed these results, in which p53 QM increased the amounts of lipin 1 in the nuclear fraction (Fig 5H). CTDNEP1 dephosphorylates and activates lipins (Han et al, 2012; Su et al, 2014). Phosphorylation levels of lipin 1 were unaffected by the presence or absence of p53 QM (Fig S3B).

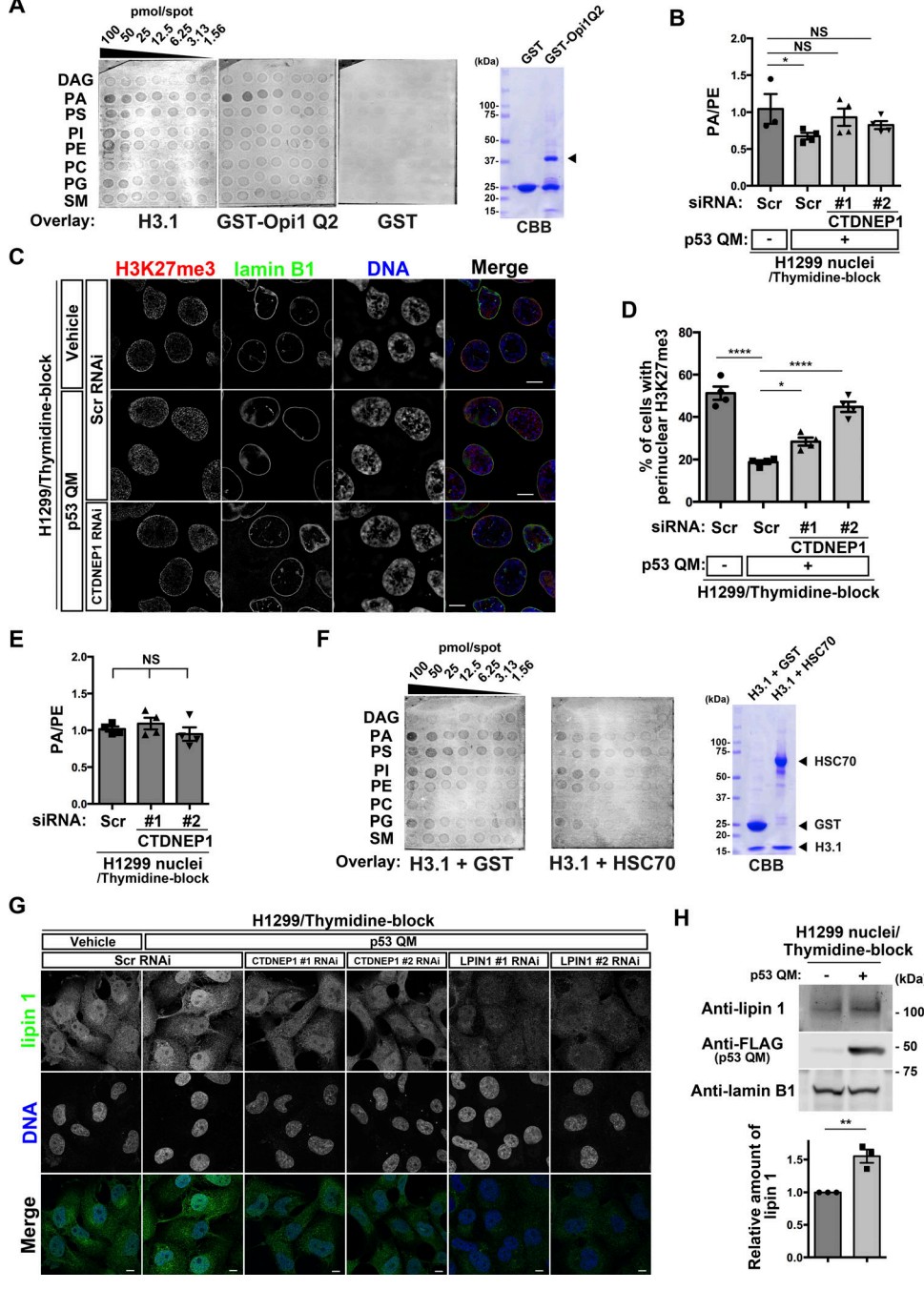

**Figure 5. p53 and C-terminal domain nuclear envelope phosphatase 1 (CTDNEP1) reduce nuclear PA levels by increasing lipin 1 in the nucleus.**
**(A)** H3.1–lipid interactions clarified by membrane lipid arrays, with GST and the Q2 domain of yeast Opi1 (Opi1 Q2) as a negative and positive control, respectively. The membrane was spotted with the indicated amounts of the following lipids: DAG, diacylglycerol; PA, phosphatidic acid; PS, phosphatidylserine; PI, phosphatidylinositol; PE, phosphatidylethanolamine; PC, phosphatidylcholine; PG, phosphatidylglycerol; and SM, sphingomyelin. CBB, Coomassie brilliant blue. The arrowhead indicates the band of GST-Opi1 Q2. The CBB blot on the right demonstrates the quality of the proteins used in this assay. **(B)** Absolute quantification of PA and PE in the nuclei of H1299 cells. PA/PE ratios are indicated. n = 3 or 4 biological replicates; *$P < 0.05$ and NS, not significant, one-way ANOVA followed by Dunnett's multiple comparisons test. **(C)** Representative structured illumination microscopy images of H1299 cells showing the localization of H3K27me3 (red), lamin B1 (green), and DNA (blue). Individual fluorescence channels are shown in grayscale with merged images on the right. Bars, 10 $\mu$m. **(D)** Quantification of H1299 cells with perinuclear H3K27me3 accumulation. n = 4 biological replicates, ≥ 100 cells per replicate; ****$P < 0.0001$ and *$P < 0.05$, one-way ANOVA followed by Dunnett's multiple comparisons test. **(E)** Absolute quantification of PA and PE in the nuclei of H1299 cells. PA/PE ratios are indicated. n = 4 biological replicates; NS, not significant, one-way ANOVA. **(F)** H3.1–lipid interactions in the presence of equimolar amounts of GST or HSC70 were clarified by membrane lipid arrays, as in (A). The CBB blot on the right demonstrates the quality of the proteins used in this assay. **(G)** Representative confocal images of H1299 cells showing the localization of lipin 1 (green) and DNA (blue). Individual fluorescence channels are shown in grayscale with merged images on the bottom. Bars, 10 $\mu$m. **(H)** Representative immunoblots of the indicated antibodies. Normalized lipin 1/lamin B1 ratios are shown. n = 3 biological replicates; **$P < 0.01$, two-tailed unpaired $t$ test.
Source data are available for this figure.

Collectively, p53 may increase CTDNEP1 levels, and CTDNEP1 may in turn increase lipin 1 and decrease nuclear PA levels, thus alleviating the abnormal accumulation of H3K27me3 near the NE. On the contrary, as antibodies against lipin 2 and lipin 3 were not available, we were unable to exclude the possible involvement of these isoforms in CTDNEP1 function.

## p53 induces *TMEM255A* to increase CTDNEP1 levels

We then sought to understand the possible molecular link between p53 and CTDNEP1. The *CTDNEP1* gene promoter does not contain

p53-binding sites, and *CTDNEP1* mRNA was not increased by p53 QM in H1299 cells (Fig S3C). We, however, found that si*TP53* reduces CTDNEP1 protein levels in MCF7 cells without reducing its mRNA (Fig S3D and E). si*CTDNEP1* also reduced lipin 1 protein levels in H1299 cells (Figs 6A and S3A).

CTDNEP1-regulatory subunit-1 (NEP1R1) was shown to increase CTDNEP1 protein levels by binding to CTDNEP1 (Han et al, 2012; Su et al, 2014; Jacquemyn et al, 2021). However, neither p53 QM nor p53 WT induced *NEP1R1* in H1299 cells (Fig 6B). We then found that a hitherto poorly characterized gene, namely, *TMEM255A* (also called *FAM70A*), was induced by p53 WT and p53 QM in H1299 cells (Fig 6C

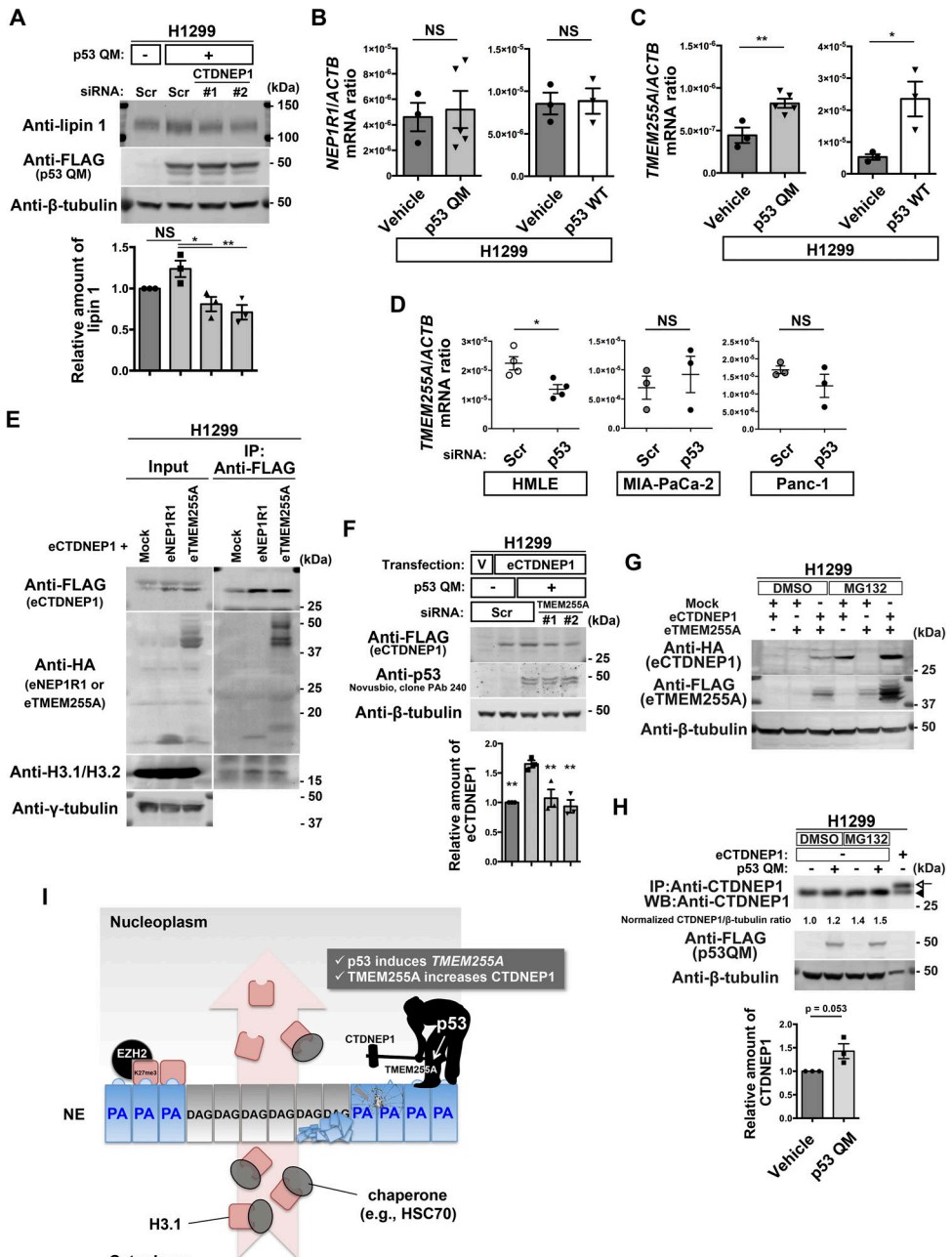

**Figure 6. TMEM255A increases C-terminal domain nuclear envelope phosphatase 1 (CTDNEP1) downstream of p53.**
**(A)** Representative immunoblots using the indicated antibodies. Normalized lipin 1/β-tubulin ratios are shown. n = 3 biological replicates; **P < 0.01, *P < 0.05, and NS, not significant, one-way ANOVA followed by Dunnett's multiple comparisons test. **(B, C)** Quantification of *NEP1R1* (B) and *TMEM255A* (C) mRNA (normalized by *ACTB* mRNA) in H1299 cells. n ≥ 3 biological replicates; **P < 0.01, *P < 0.03, and NS, not significant, two-tailed unpaired *t* test. **(D)** Quantification of *TMEM255A* mRNA (normalized by *ACTB* mRNA). n ≥ 3 biological replicates; *P < 0.02 and NS, not significant, two-tailed unpaired *t* test. **(E)** Representative immunoblots using the indicated antibodies. IP, immunoprecipitation. **(F)** Representative immunoblots using the indicated antibodies. V, vehicle. Normalized eCTDNEP1/β-tubulin ratios are shown. n = 3 biological replicates; **P < 0.01, one-way ANOVA followed by Dunnett's multiple comparisons test. **(G)** Representative immunoblots using the indicated antibodies. **(H)** Representative immunoblots using the indicated antibodies. IP, immunoprecipitation. WB, Western blotting. The arrowhead indicates endogenous CTDNEP1, and the arrow indicates eCTDNEP1. Normalized CTDNEP1/β-tubulin ratios of the DMSO-treated samples are shown at the bottom. n = 3 biological replicates; P = 0.053, two-tailed unpaired *t* test. **(I)** Model for the inhibition of the perinuclear accumulation of H3K27me3 regulated by p53.
Source data are available for this figure.

and Table S1). *TMEM255A* mRNA was also induced by endogenous p53 WT in HMLE cells (Figs 6D and S3F), although we have not yet clarified whether the *TMEM255A* gene is a direct target of p53. On the contrary, *TMEM255A* mRNA was not induced by mutant p53s expressed in cancer cells, such as MIA-PaCa-2 cells (expressing p53 R248W) and Panc-1 cells (expressing p53 R273H) (Figs 6D and S3F), in which the perinuclear accumulation of H3K27me3 was observed (Fig S3F and G).

We then hypothesized that the TMEM255A protein increases CTDNEP1 protein levels. The expression of TMEM255A increased CTDNEP1 levels in reconstitution experiments, in which we expressed cDNAs encoding epitope-tagged TMEM255A (eTMEM255A:

TMEM255A-Myc-FLAG or HA-TMEM255A) and epitope-tagged CTDNEP1 (eCTDNEP1: CTDNEP1-Myc-FLAG or HA-CTDNEP1) (Figs 6E and S3H). Moreover, eTMEM255A was clearly coprecipitated with eCTDNEP1 (Figs 6E and S3H). Furthermore, p53 QM increased eCTDNEP1 levels in H1299 cells, and si*TMEM255A* canceled this increase (Figs 6F and S3I). Furthermore, the single expression of either of these cDNAs did not lead to the robust expression of the corresponding protein (Fig 6G). Therefore, CTDNEP1 appears to require TMEM255A for its protein expression. Supporting this notion, ColabFold analysis of the predicted 3D structure of protein complexes (Mirdita et al, 2022) suggested that CTDNEP1 can make a complex with TMEM255A through its transmembrane alpha helix

(Fig S3J), although we have yet to confirm this possibility bio-chemically. Moreover, inhibition of proteasome activity by MG132 increased protein levels of both eCTDNEP1 and eTMEM255A in H1299 cells (Fig 6G), and the level of endogenous CTDNEP1 was increased in the presence of p53 QM and MG132 (Fig 6H). Therefore, it is likely that p53 induces the *TMEM255A* gene, and its protein product may stabilize CTDNEP1 via circumventing protein degradation, possibly by their association.

On the contrary, the degree of CTDNEP1 increase by the p53-TMEM255A-CTDNEP1 axis appeared to be smaller than what we observed earlier in the H3.1 interactome. We also found substantial changes in the subcellular localization of HA-CTDNEP1; a higher amount of HA-CTDNEP1 was found to be localized around the NE in the presence of p53 QM than in its absence (Fig S3K). Thus, mechanisms other than this might also exist by which p53 increases CTDNEP1 in the H3.1 interactome.

### p53 triggers the expression of H3K27me3-marked genes in H1299 cells

Whereas our RNA-Seq analysis (Table S1) suggested that the expression of p53 QM either up-regulates or down-regulates several hundred genes by more than twofold in H1299 cells, how this mutant p53 modulates gene expression is unknown. We then extracted these genes and investigated their overlaps with the curated 5,501 gene sets available in the Molecular Signatures Database. Interestingly, we found that genes marked by H3K27me3 at their promoters in various cells, such as human embryonic stem cells or MEFs, significantly overlapped with the up-regulated genes (Table S3). Furthermore, genes identified as SUZ12 or embryonic ectoderm development targets in human embryonic stem cells also significantly overlapped with the up-regulated genes (Table S3). On the contrary, none of the gene sets significantly overlapped with the down-regulated genes (Table S3). Given that most of the H3K27me3 and PRC2 targets are repressed genes, the above results suggest that p53 QM canceled their repression. We further analyzed the expression levels of some of the up-regulated genes that were suggested to be H3K27me3 or PRC2 targets in the Molecular Signatures Database. We found that NK2 homeobox 3 (*NKX2-3*) was up-regulated upon the expression of p53 QM in H1299 cells (Fig S4A). We also found that the levels of H3K27me3 at its promoter region are reduced upon p53 QM expression, whereas the levels of H3K27ac and H3K4me3 tend to be increased (Fig S4B). Therefore, the peri-nuclearly tethered H3K27me3 found in the absence of p53 may add a stratum of epigenetic regulation at some gene loci.

## Discussion

Our results described in this study demonstrate a novel function of p53 in the regulation of nuclear lipid components, and show that this function is crucial to normalize the nuclear behavior of H3.1. We have also identified an unexpected function of EZH2 that may emerge when the above function of p53 is impaired. As a result, in the absence of p53, H3.1 molecules may be tethered around the NE

and therein be marked as suppressive by EZH2 without forming nucleosomes during the G1/S phase of the cell cycle (Fig 6I).

One of the major targets of p53 in normalizing the nuclear behavior of H3.1 is the down-regulation of nuclear PA levels. CTDNEP1 is primarily responsible for this down-regulation, and its amounts are increased in the H3.1 interactome by p53. p53 induces *TMEM255A* gene expression, and its protein product then stabilizes the CTDNEP1 protein. Whether the *TMEM255A* promoter is a direct target of p53 should be clarified in the future study, as p53 was shown to bind directly to the *TMEM255A* locus in mouse embryonic fibroblasts (Kenzelmann Broz et al, 2013). Another action of p53 is the down-regulation of EZH2 within the H3.1 interactome, and p53 has previously been shown to suppress the *EZH2* gene promoter (Tang et al, 2004). Thus, the duality of p53 in transcriptional regulation may underlie the herein identified function of p53. However, many issues remain unsolved. For example, the p53-TMEM255A-CTDNEP1 axis may not fully explain the p53-mediated increase in CTDNEP1 levels in the H3.1 interactome, as mentioned earlier. Total cellular EZH2 protein levels were not notably changed in H1299 cells by p53 (see Fig 4A), and hence, some unknown mechanisms might exist that specifically regulate the amounts of EZH2 in the H3.1 interactome. Mechanisms by which p53 uses CTDNEP1 in the H3.1 interactome also await to be elucidated.

In the absence of p53, H3.1 molecules may be tethered at or be located near the NE after they have entered the nucleus, rather than be directly trapped at the NE during their entry into the nucleus. H3.1 shows robust affinity to PA, whereas its chaperone HSC70 can attenuate the H3.1-PA association. HSC70 is thought to be associated with H3.1 primarily in the cytoplasm (i.e., thus originally named a "cytoplasmic" chaperone), and most nuclear H3.1 molecules were shown to be associated with other chaperones, such as CAF1 (Campos et al, 2010). Therefore, H3.1 molecules appear to be dissociated from HSC70 after entering the nucleus, and may use other chaperones, such as CAF1, in the nucleus. Taken together, these properties would explain to some extent why H3.1 goes through the NE and then becomes anchored around the NE in the absence of p53. Mechanisms regulating PA in the NE and in the nuclear pore complex have been extensively studied (Penfield et al, 2020; Jacquemyn et al, 2021; Merta et al, 2021; Thaller et al, 2021; Barger et al, 2022), because of their possible association with genetic/epigenetic control and integrity. Hence, understanding the detailed mechanisms by which PA constrains H3.1 molecules at the NE, and also the detailed mechanisms as to how EZH2 targets such H3.1 molecules being anchored to the NE without forming nucleosomes will be of particular importance. Whether such aberrantly marked H3.1 molecules will then be discarded from the nucleus, or be used in some way in the nucleosomes should also be clarified.

Given that the PRC2-targeted or H3K27me3-marked genome regions are typically replicated in the late phases of DNA replication in fly cells (Lo Sardo et al, 2013) or human cancer cells (Du et al, 2019), we envision that monomeric H3K27me3 anchored to the NE can be incorporated into the PRC2 regions. It was shown that H3K27me3 spreads over neighboring nucleosomes via the read-and-write mechanism, in which embryonic ectoderm development first recognizes H3K27me3 and then allosterically activates EZH2 to stimulate the further catalysis of H3K27me3 (Hansen et al, 2008; Margueron et al, 2009; Reinberg & Vales, 2018). Therefore, the PRC2 regions in daughter cells might contain an

increased amount of H3K27me3 to be transcriptionally suppressed when p53 is absent (Table S3 and Fig S4). This mechanism might be beneficial to cells lacking H3.1 management by p53, as a repressive and heritable histone modification that precedes nucleosome formation would securely avoid uncontrollable gene expression over multiple generations (Reinberg & Vales, 2018). Reveron-Gomez et al demonstrated that a substantial number of H3K27me3 regions in the HeLa cell genome show a relatively faster rate of restoration after being diluted by DNA replication (Reveron-Gomez et al, 2018). HeLa cells express the E6 protein, which is derived from an endogenous papillomavirus, and hence, the p53 protein is undetectable in these cells by Western blotting, although they harbor an intact *TP53* gene (Matlashewski et al, 1986; Scheffner et al, 1990). Therefore, the accumulation of H3K27me3 in the NE demonstrated in our study might serve as a nucleation site for this faster H3K27me3 propagation at some gene loci for effective gene repression.

We have previously shown that the loss of normal p53 in epithelial cells may cause the aberrant deposition of H3K27me3 on epithelial-specific gene loci, such as *CDH1*, and that p53 uses EZH2 in this process (Oikawa et al, 2018a, 2018b). However, the aberrant accumulation of H3K27me3 also occurs in p53-deficient mouse fibroblasts. Thus, the function of p53 in controlling the normal behavior of H3.1 does not appear to be specific to epithelial cells.

In summary, we found that there is a tight control on the nuclear behavior of H3.1 by p53, and demonstrated unprecedented perspectives regarding p53 function and EZH2-mediated H3K27me3 modification that may occur with unmethylated H3.1 molecules at or near the NE. Then, an outstanding question is if such functions of p53 and EZH2 exist, whether any of them are associated with epigenetic control and integrity. Whether such regulations are necessary only for the G1/S-specific histones should also be clarified.

# Materials and Methods

## Plasmid construction and retroviral gene transduction

A cDNA encoding a p53 protein with quadruple mutations (L22Q/W23S/W53Q/F54S; p53 QM) was generated by PCR-based mutagenesis (Hashimoto et al, 2016). The cDNA for p53 WT was cloned into pRetroX-IRES-ZsGreen1 (Clontech). The cDNA for p53 WT or p53 QM was cloned together with the DNA sequence for an NH$_2$-terminal FLAG tag into pRetroX-Tight-Pur (Clontech). Retroviruses with the vesicular stomatitis virus G (VSV-G) envelope were produced by transfection of GP2-293 cells (Clontech) with the pRetroX construct and pVSV-G (Clontech) using Lipofectamine LTX reagent (Invitrogen), following the manufacturer's instructions. cDNAs encoding *CTDNEP1*, *NEP1R1*, and *TMEM255A* followed by the Myc-FLAG tag were purchased from OriGene (#RC203657, #RC235833, and #RC207956, respectively). To produce the Q2 domain of the Opi1 protein tagged with GST, a cDNA encoding the Q2 domain of Opi1 (a gift from Dr. Alwin Köhler, University of Vienna, Austria) was cloned into pGEX-6P-1 (Amersham Pharmacia Biotech).

## Cell culture

H1299 cells, A549 cells, MCF7 cells, MB352 cells, MIA-PaCa-2 cells, and Panc-1 cells were obtained from the American Type Culture Collection, and cultured under 5% CO$_2$ at 37°C in DMEM supplemented with 10% FBS. HMLE cells were generated by introducing SV40 large T-antigen and human telomerase reverse transcriptase into a primary culture of normal mammary epithelial cells (Lonza). HMLE cells were cultured in Mammary Epithelial Cell Growth Medium (Lonza). No mycoplasmas were detected in cultures by DAPI or TO-PRO-3 (Thermo Fisher Scientific) staining. Polyclonal cell lines capable of the inducible expression of normal p53 (p53 WT) or a mutant p53 (p53 QM) were generated by infection of H1299 cells with the corresponding retrovirus and the rtTA retrovirus (Clontech), followed by selection with puromycin (1 μg/ml) and geneticin (G418, 500 μg/ml). The cells were cultured for 48–72 h in the presence of doxycycline (0.5 μg/ml) to induce the corresponding genes. Polyclonal cell lines expressing an epitope-tagged CTDNEP1 (MCF7-eCTDNEP1) were generated by the transfection of MCF7 cells with a vector encoding CTDNEP1, followed by selection with geneticin (G418, 250 μg/ml). H1299 and MB352 cells were stably transfected with pPB vectors encoding Fucci probes (Onodera et al, 2018).

## Antibodies and reagents

Primary antibodies for immunofluorescence, PLA, and immunoblot analysis were purchased from commercial sources, as listed in Table S4. Thymidine (#T9250; Sigma-Aldrich) was used to inhibit S-phase progression. DSP (#D629; Dojindo) was used to crosslink cellular proteins at 1 mM for 30 min at RT. Bis(sulfosuccinimidyl) suberate disodium salt (BS3; #B574; Dojindo) was used to crosslink the antibodies to protein G magnetic beads (Cell Signaling Technology). Benzonase (#70664-3CN; Merck) was added to the nuclear extracts at 100 units/ml with 1 mM MgCl$_2$ overnight at 4°C to digest nucleic acids. Deoxycytidine (#D3897; Sigma-Aldrich) was used to release cells from the thymidine-mediated blockade of cell-cycle progression. MG132 (#BML-PI102-0005; Enzo Life Sciences) was used at 1 μM for 20 h to inhibit proteasomal function.

## Western blotting

Samples were loaded onto e-PAGEL gel (ATTO) or TGX FastCast gel (Bio-Rad), and electrophoresed, and then, the proteins in the gel were transferred onto Immobilon-FL polyvinylidene difluoride membranes (Millipore). The membranes were blocked with BlockPRO Protein-free Blocking Buffer (Visual Protein) or 1% BSA and 5% skim milk in PBS, and then incubated with primary antibody solution at 4°C overnight. After washing with PBS containing 0.1% Tween-20 (PBST), the membrane was incubated with IRDye secondary antibodies (LI-COR) or horseradish peroxidase–conjugated secondary antibodies (Jackson ImmunoResearch Laboratories) at RT for 1 h. To detect the immunoprecipitated endogenous CTDNEP1, TidyBlot horseradish peroxidase–conjugated secondary antibody (Bio-Rad) was used to enable the detection of immunoblotted target protein bands without interference from denatured IgG. After

washing again with PBST, the membrane was scanned with Odyssey Imaging System (LI-COR) or detected with ImageQuant LAS 4000 mini imager (GE Healthcare), followed by its reaction with detection reagents (ECL Western Blotting Detection Reagents; GE Healthcare, or SuperSignal West Dura; Thermo Fisher Scientific). The primary antibodies used and their dilutions are listed in Table S4. All images were processed using ImageJ and/or Photoshop software (Adobe).

## Cell-cycle analysis

H1299 cells transduced with a vector encoding p53 WT or p53 QM were cultured with or without thymidine for 20 h. Cells were then trypsinized, washed twice in PBS, and fixed with ice-cold 70% ethanol for 2 h. Fixed cells were then washed twice in PBS, and incubated with 0.5 mg/ml RNase A (#R4642; Sigma-Aldrich) at 37°C for 30 min, followed by staining of DNA with 50 µg/ml propidium iodide. Stained cells were subjected to flow cytometric analysis using FACSVerse (BD).

## Immunofluorescence analysis

For immunofluorescence analysis, cells cultured on coverslips were fixed either with 4% PFA in PBS for 10–20 min followed by permeabilization with 0.1% Triton X-100 in PBS for 10 min, or with 100% ethanol for 10 min at −20°C, and incubated with blocking solution (1% BSA in PBS) for more than 60 min at RT. The cells were then incubated with primary antibodies for 60 min at RT, then washed with blocking solution, and incubated with Alexa Fluor 488–, Alexa Fluor 647–, or Cy3-conjugated secondary antibodies (Molecular Probes) for 30 min. The cells were also stained with TO-PRO-3 dye or DAPI to visualize nuclei. The cells were finally washed with blocking solution, mounted onto glass slides with ProLong Diamond antifade reagent (Thermo Fisher Scientific), observed using a confocal laser scanning microscope with an oil-immersion objective (CFI Plan Apo VC 100×/1.4 NA or Apo λS 60×/1.4 NA), and analyzed with the attached software (Model A1 or A1R with NIS-Elements, Nikon), or with an N-SIM microscope (Nikon) with an oil-immersion objective (100×/1.49 NA), laser illumination, and an electron-multiplying charged-coupled device camera. Image reconstruction was performed using NIS-Elements software (Nikon). All images were processed using Photoshop software.

## Quantification of the perinuclear accumulation of H3K27me3

Each nuclear compartment was divided into three equally spaced concentric elliptic regions, starting from the boundary of the nucleus (selected manually using NIS-Elements software). The outermost concentric regions were taken as the perinuclear regions, and the innermost concentric regions were taken as the central nuclear regions. The average pixel intensities of H3K27me3 in these regions were measured by NIS-Elements software. The perinuclear accumulation of H3K27me3 was defined as their ratio (i.e., the average H3K27me3 intensity of the perinuclear regions/the central nuclear regions) being more than 2. Nuclei with aberrant morphology (i.e., nuclei to which ellipses do not fit) were omitted from this analysis (Fig S1A).

## PLA

To visualize the subcellular localization of the H3K27me3-lamin A/C, H3K27me3-H4, H3.1/H3.2-lamin A/C, and H3-EZH2 interactions, H1299 cells with or without p53 QM expression were cultured with thymidine for 20 h to be arrested in the S phase. Cells were then fixed, permeabilized, and incubated with primary antibodies, as described in the immunofluorescence analysis section. PLA reactions were performed using Duolink In Situ PLA Probe Anti-Mouse PLUS, Anti-Rabbit MINUS, and Detection Reagents Orange (Sigma-Aldrich), following the manufacturer's instructions. After the ligation and amplification reactions, cells were further incubated with the Alexa Fluor 488–conjugated anti-lamin B1 antibody (#sc-365214AF488; Santa Cruz) and TO-PRO-3 dye to visualize the nuclear lamina and DNA, respectively. Specificities of the antibodies used for the PLA were confirmed by immunostaining performed at the same concentrations.

## Cell-cycle synchronization and release

Cells were synchronized at the G1/S border or the S phase by a single thymidine block (2 mM thymidine, 18–20 h) and then released by incubation in fresh media containing deoxycytidine (24 µM).

## Labeling of newly synthesized DNA

To label and visualize newly replicating DNA, the cells released from the thymidine block by incubation in fresh media containing deoxycytidine (24 µM) for 2 h were quickly rinsed with a hypotonic buffer (10 mM Hepes [pH 7.4] and 50 mM KCl), and incubated in this buffer containing 0.1 mM biotin–dUTP (#11093070910; Roche) for 10 min. Cells were then cultured in fresh culture medium for 20 min before fixation. Biotin–dUTP incorporated into the DNA replication sites were visualized by Alexa Fluor 488–conjugated streptavidin (#S32354; Thermo Fisher Scientific).

## Protein crosslinking and immunoprecipitation

For crosslinking, cells were washed with PBS and incubated with 1 mM DSP in PBS for 30 min at RT. Cells were then lysed with cell lysis buffer (10 mM Hepes [pH 7.4], 10 mM KCl, 0.05% NP-40) supplemented with a protease inhibitor cocktail (#4693116001; Merck), and incubated on ice for 20 min. The lysates were then centrifuged at 15,600g for 10 min at 4°C. The supernatants (cytosolic fraction) were removed, and the pellets (nuclei) were resuspended in modified RIPA buffer (50 mM Tris [pH 7.4], 500 mM NaCl, 1 mM EDTA, 1% Triton X-100, 1% SDS, and 1% sodium deoxycholate) supplemented with the protease inhibitor cocktail, bath-sonicated, and incubated on ice for 20 min. The lysates were then centrifuged at 17,900g for 10 min at 4°C. The supernatants (crosslinked nuclear extracts) were diluted to yield 140 mM NaCl and 0.1% SDS solutions for immunoprecipitation. For immunoprecipitation, normal rabbit IgG or anti-H3 antibodies were crosslinked to protein G magnetic beads by BS3. The crosslinked nuclear extracts were then immunoprecipitated with antibody-conjugated beads overnight at 4°C. Samples for MS analysis or Western blot analysis were prepared by incubating and boiling the beads in SDS sample buffer.

## Identification of H3.1-interacting proteins by liquid chromatography–tandem MS (LC-MS/MS)

Samples were reduced with 10 mM TCEP at 100°C for 10 min, alkylated with 50 mM iodoacetamide at an ambient temperature for 45 min, and then subjected to SDS–PAGE. Electrophoresis was stopped at a migration distance of 2 mm from the top edge of the separation gel. After Coomassie brilliant blue staining, protein bands were excised, destained, and cut finely before in-gel digestion with trypsin/Lys-C mix (Promega) at 37°C for 12 h. The resulting peptides were extracted from gel fragments and analyzed using the Orbitrap Fusion Lumos mass spectrometer (Thermo Fisher Scientific) combined with UltiMate 3000 RSLC nano-flow HPLC (Thermo Fisher Scientific). Peptides were enriched with $\mu$-Precolumn (0.3 mm i.d. × 5 mm, 5 $\mu$m; Thermo Fisher Scientific) and separated on an AURORA column (0.075 mm i.d. × 250 mm, 1.6 $\mu$m; Ion Opticks Pty Ltd.) using the two-step gradient: 2–40% acetonitrile for 110 min, followed by 40–95% acetonitrile for 5 min in the presence of 0.1% formic acid. The analytical parameters of Orbitrap Fusion Lumos were set as follows: resolution of full scans = 50,000; scan range (m/z) = 350 to 1,500; maximum injection time of full scans = 50 msec; AGC target of full scans = 4 × $10^5$; dynamic exclusion duration = 30 s; cycle time of data-dependent MS/MS acquisition = 2 s; activation type = HCD; detector of MS/MS = ion trap; maximum injection time of MS/MS = 35 msec; AGC target of MS/MS = 1 × $10^4$. The MS/MS spectra were searched against a *Homo sapiens* protein sequence database (20,366 entries) in SwissProt using Proteome Discoverer 2.4 software (Thermo Fisher Scientific), in which peptide identification filters were set at a false discovery rate of less than 1%. Label-free relative quantification analysis for proteins was performed with the default parameters of Minora Feature Detector node, Feature Mapper node, and Precursor Ions Quantifier node in Proteome Discoverer 2.4 software.

## MMEJ-assisted gene knock-in using CRISPR/Cas9 with the precise integration into target chromosome (PITCh) system

For stable genomic integration of Dronpa with puromycin N-acetyltransferase or blasticidin deaminase into the carboxy terminus of H3.1, H1299 cells were transfected with two plasmids. The first plasmid encodes the inserted sequence flanked by two microhomology arms with a length of 40 bp and containing short PITCh sequences at their distal ends (modified based on pCRIS-PITChv2-FBL, #63672; Addgene) (Sakuma et al, 2016). The second plasmid directs the expression of appropriate sgRNAs for cleaving the PITCh sequences and for targeting of the *HIST1H3A* locus to create a DNA fragment and Cas9 (modified based on pX330A-1x2, #58766; Addgene, and pX330S-2-PITCh, #63670; Addgene) (Sakuma et al, 2016). From 3 d after transfection, the amount of puromycin (up to 0.7 $\mu$g/ml) or blasticidin (up to 7 $\mu$g/ml) was increased gradually for 20 d to enable the proliferation of cells with genomic insertion of the resistance gene.

## FRAP analysis of H3.1-Dronpa

To analyze nuclear recovery of the H3.1-Dronpa signals, their fluorescence in the nucleus was fully photobleached for 0.5 s using the 488-nm laser at its maximum (100%) intensity. Immediately after photobleaching, a 2- to 3-h time series was acquired taking 1 image every 5 min. The intensity of the 488-nm laser was at 0.1–0.4% of the maximum, to minimize bleaching of the signal.

## Protein lipid overlay assay

Nitrocellulose membranes spotted with serial dilutions of different lipids (Membrane Lipid Arrays, #P-6003; Echelon Biosciences) were used to assess the lipid binding activity of histone H3.1. Membranes were incubated with blocking buffer (3% fatty acid–free BSA [#017-15141; Fujifilm] and 0.1% Tween-20 in PBS) for 1 h at RT before being incubated with 1.0 $\mu$g/ml of recombinant human histone H3.1 (#M2503S; New England Biolabs) or GST-fusion proteins for 1 h at RT. Alternatively, 1.0 $\mu$g/ml of recombinant human histone H3.1 was mixed with an equimolar amount of GST or recombinant human HSC70 (#ab78431; Abcam). The membrane was then washed with 0.1% Tween-20 in PBS, and incubated with an anti-histone H3.1/H3.2 antibody or anti-GST antibody for detection by immunoblotting.

## Phosphatase assay

To analyze the phosphorylation status of lipin 1, cells were lysed with 2× PPase reaction buffer (1× buffer contains 50 mM Hepes [pH 7.5], 100 mM NaCl, 10% glycerol, and 0.5% Triton X-100) supplemented with a protease inhibitor cocktail (#4693116001; Merck). The lysates were then bath-sonicated on ice for 5 min, and centrifuged at 15,600*g* for 10 min at 4°C. Total lysates (200 $\mu$g each) were incubated with or without 400 U lambda protein phosphatase (#P0753S; New England Biolabs) and 1 mM $MnCl_2$ at 30°C for 30 min. The reaction was stopped with 5× SDS sample buffer, and boiled for 5 min before Western blot analysis.

## Quantification of PA in H1299 nuclei by LC-MS/MS

To prepare H1299 cell nuclei, cells were collected and washed with ice-cold PBS on ice. Cells were lysed with a hypotonic buffer (10 mM Hepes [pH 7.4], 1.5 mM $MgCl_2$, and 10 mM KCl) supplemented with 1 mM DTT, a protease inhibitor cocktail (#4693116001; Merck), and a phosphatase inhibitor cocktail (#04906837001; Sigma-Aldrich), and then incubated on ice for 15 min. NP-40 at 0.5% (vol/vol) was then added to the lysates, and the lysates were briefly vortexed and centrifuged at 10,000*g* for 40 s at 4°C. The supernatant (cytosolic fraction) was removed, and the pellet (nuclei) was resuspended in the hypotonic buffer (10 mM Hepes [pH 7.4], 1.5 mM $MgCl_2$, and 10 mM KCl) supplemented with 1 mM DTT, the protease inhibitor cocktail, and the phosphatase inhibitor cocktail. The nuclei were then centrifuged at 10,000*g* for 40 s at 4°C, washed twice with ice-cold PBS, and then subjected to lipid extraction.

A 50 $\mu$l methanol solution containing 20 pmol each of C15:0/18:1 PA (#330721; Avanti Polar Lipids) and C12:0/13:0 PE (#LM1100; Avanti Polar Lipids) was added to each nuclear sample in 1.2 ml methanol in a glass tube, followed by the addition of ultrapure water (400 $\mu$l), 2 M HCl (400 $\mu$l), and 1 M NaCl (200 $\mu$l). After vigorous vortex mixing, 2 ml $CHCl_3$ was added, followed by further vortexing for 1 min. After centrifugation (1,200*g* for 3 min at RT), the lower organic phase (crude lipid extract) was collected and transferred to a new glass

tube. Purified phospholipids were derivatized by methylation using a method described previously (Morioka et al, 2022). Briefly, 150 μl of 0.6 M (trimethylsilyl)diazomethane (#T1146; Tokyo Chemical Industry) was added to the purified phospholipid fraction prepared as above at RT. After 10 min, the reaction was quenched with 10 μl glacial acetic acid. The samples were mixed with 400 μl CHCl$_3$ and 400 μl ultrapure water, followed by vortexing for 1 min. After centrifugation at 1,200$g$ for 3 min, the lower phase was dried under a stream of nitrogen, and redissolved in 100 μl acetonitrile.

LC-MS/MS was performed using a triple quadrupole mass spectrometer QTRAP 6500 (AB Sciex) and a Nexera X2 HPLC system (Shimadzu) combined with a PAL HTC-xt autosampler (CTC Analytics). The mass range of the instrument was set at 5–2,000 m/z. Spectra were recorded in the positive ion mode as [M+H]$^+$ ions for the detection of PAs and PEs. The ion spray voltage was set at 5.5 kV, cone voltage at 30 V, and source block temperature at 100°C. Curtain gas was set at 20 psi, collision gas at 9 psi, ion source gas pressures 1/2 at 50 psi, declustering potential at 100 V, entrance potential at 10 V, collision cell exit potential at 12 V, and collision energy values at 35 eV. Lipid samples (10 μl each) dissolved in acetonitrile were injected using the autosampler, and molecules were separated using a Bio C18 column (1.9 μm, 2.1 × 150 mm) at 60°C. LC was operated at a flow rate of 100 μl/min with a gradient as follows: 90% mobile phase A (methanol/acetonitrile/deionized water = 18/18/4 containing 5 mM ammonium acetate) and 10% mobile phase B (2-propanol containing 5 mM ammonium acetate) were maintained for 2 min, linearly increased to 82% mobile phase B over 17 min, and maintained at 85% mobile phase B for 4 min. The column was re-equilibrated to 10% mobile phase B for 10 min before the next injection. Data were analyzed by MultiQuant 3.0.2 software (AB Sciex). Concentrations of each molecular species were calculated by dividing the peak area of each species by the peak area of the corresponding internal surrogate standard. The sum of each phospholipid molecular species was calculated, and the ratio (PA/PE) was presented in the figure.

## RNA interference (RNAi) experiments

Silencer Select or Silencer Pre-designed siRNAs targeting coding sequences of human *TP53* mRNA (5′-GUAAUCUACUGGGACGGAATT-3′: p53 #1, and 5′-GGUGAACCUUAGUACCUAATT-3′: p53 #2), human *CTDNEP1* mRNA (5′-GUACCAAACUGUUCGAUAUTT-3′: CTDNEP1 #1, and 5′-GGAUCUGGAUGAGACACUUTT-3′: CTDNEP1 #2), human *LPIN1* mRNA (5′-GACUUUCCCUGUUCGGAUATT-3′: LPIN1 #1, and 5′-GGA-GUGUCUUUGAAUAGAATT-3′: LPIN1 #2), human *TMEM255A* mRNA (5′-AAACCUUAUUGGAGAACAAATT-3′: TMEM255A #1, and 5′-GGCUUUAAG-GACAUGAACCTT-3′: TMEM255A #2), or Stealth siRNAs targeting coding sequences of human *EZH2* mRNA (5′-GACCACAGU-GUUACCAGCAUUUGGA-3′: EZH2 #1, and 5′-GAGCAAAGCUUACA-CUCCUUUCAUA-3′: EZH2 #2) were obtained from Thermo Fisher Scientific. Silencer Select negative control siRNA and Stealth negative control siRNA were purchased from Thermo Fisher Scientific. For the transfection of siRNA, cells were plated at 15–20% confluence in six-well plates together with 15 nM Silencer Select or Silencer Pre-designed siRNA and 3.5 μl of RNAi MAX (Thermo Fisher Scientific), or with 150 pmol of Stealth siRNA and 3.5 μl of RNAi MAX.

They were then cultured in complete medium for 24 h before a second round of siRNA transfection.

## Quantitative reverse transcription (RT)–PCR analysis

Total RNA was extracted from cells using TRIzol reagent (Thermo Fisher Scientific) and purified with Direct-zol RNA MiniPrep Kit (Zymo Research), and aliquots (1 μg) of the RNA were subjected to RT with SuperScript IV polymerase (Thermo Fisher Scientific). TaqMan RT–PCR primers for *NEP1R1*, *TMEM255A*, *CTDNEP1*, *NKX2-3*, and *ACTB* were obtained from Applied Biosystems for quantitative PCR analysis. Quantitative PCR analysis was performed using the 7300 Fast Real-Time PCR System (Applied Biosystems). For the quantification of *NEP1R1*, *TMEM255A*, *CTDNEP1*, and *ACTB*, absolute quantification was performed using plasmids encoding human *NEP1R1*, *TMEM255A*, *CTDNEP1*, and *ACTB* as a standard, respectively. For the quantification of *NKX2-3*, delta Ct values, which were the Ct values of *ACTB* subtracted from the Ct values of each gene, were used for normalization, and relative mRNA levels were calculated by $2^{-\Delta\Delta Ct}$.

## ChIP–quantitative PCR (ChIP–qPCR) analysis

ChIP was performed using Simple ChIP Enzymatic Chromatin IP Kit (Cell Signaling Technology) according to the manufacturer's instructions. The immunoprecipitated DNA was evaluated by qPCR. Sequences of the qPCR primers used are as follows for quantification of the promoter region of *NKX2-3*: 5′-AGCGAGCCCAAGGAA-CATG-3′ (forward), and 5′-GCGGGTGCGTTTTCTTTCC-3′ (reverse), with FAM- and TAMRA-labeled 5′-CTCACTTTGGCTCCGGTCCCTCACGA-3′ as the probe.

## RNA-Seq for H1299 cells

Total RNA was isolated from each sample using Direct-zol RNA MiniPrep Kit (Zymo Research) together with DNase I treatment. The integrity and quantity of the total RNA was measured using Agilent 2100 Bioanalyzer (RNA 6000 Nano Kit; Agilent Technologies). Total RNA obtained from each sample was subjected to sequencing library construction using NEBNext Ultra Directional RNA Library Prep Kit for Illumina (New England Biolabs) with NEBNext Poly(A) mRNA Magnetic Isolation Module, according to the manufacturer's protocol. The quality of the libraries was assessed using Agilent 2100 Bioanalyzer High Sensitivity DNA Kit (Agilent Technologies). The pooled libraries of the samples were sequenced using NextSeq 500 (Illumina) in 76–base pair single-end reads. Sequencing adaptors, low-quality reads, and bases were trimmed with the Trimmomatic-0.32 tool (Bolger et al, 2014). The sequence reads were aligned to the human reference genome (hg19) using TopHat 2.1.1 (bowtie2-3.2.0) (Langmead & Salzberg, 2012), which can adequately align reads onto the location including splice sites in the genome sequence. Files of the gene model annotations and known transcripts, which are necessary for whole transcriptome alignment with TopHat, were downloaded from Illumina's iGenomes website (http://support.illumina.com/sequencing/sequencing_software/igenome.html). The aligned reads were subjected to downstream analyses using StrandNGS 3.2 software (Agilent Technologies). The read counts

allocated for each gene, and transcripts (RefSeq Genes 2015.10.05) were quantified using the trimmed mean of the M-value method (Robinson & Oshlack, 2010).

## Statistical analysis

Statistical analysis was performed using GraphPad Prism 6 software (GraphPad Software, Inc.). Quantitative data are presented as the mean ± SEM. The sample size for each experiment and the replicate number of experiments are indicated in the figures or figure legends. Comparisons between two groups or more than two groups were performed using the *t* test or one-way ANOVA followed by Dunnett's multiple comparisons test, respectively. Statistical significance was indicated in the figures or figure legends. A *P*-value less than 0.05 was considered to indicate a statistically significant difference between groups.

## Supplementary Information

## Acknowledgements

We thank HA Popiel for her critical reading of the article. We also thank Dr. Alwin Köhler and Dr. Anete Romanauska (University of Vienna, Austria) for providing the cDNA encoding the Q2 domain of Opi1. We would like to thank the Nikon Imaging Center at Hokkaido University for technical support. This work was supported by JSPS KAKENHI grant numbers JP24K09447, JP20K07305, JP20H05523, and JP16H06280 (to T Oikawa); JP18H02608 and JP21H02675 (to H Sabe); JP20H03206 (to J Hasegawa); JP20H04920 (to J Sasaki); JP20H03433 (to T Sasaki); by SGH Cancer Research Grant (to T Oikawa); by AMED grant 18gm0710002h0006 (to T Sasaki); by Multilayered Stress Diseases (JPMXP1323015483), Tokyo Medical and Dental University (to T Sasaki); and by Nanken-Kyoten, Tokyo Medical and Dental University (to T Oikawa).

## Author Contributions

T Oikawa: conceptualization, resources, data curation, formal analysis, funding acquisition, validation, investigation, visualization, methodology, project administration, and writing—original draft, review, and editing.

J Hasegawa: resources, data curation, formal analysis, funding acquisition, validation, investigation, methodology, and writing—original draft, review, and editing.

H Handa: resources, validation, investigation, and methodology.

N Ohnishi: resources, data curation, formal analysis, validation, investigation, and methodology.

Y Onodera: resources and methodology.

A Hashimoto: data curation, investigation, and methodology.

J Sasaki: resources, data curation, formal analysis, funding acquisition, validation, investigation, methodology, and writing—original draft.

T Sasaki: resources, data curation, formal analysis, funding acquisition, validation, investigation, methodology, project administration, and writing—original draft.

K Ueda: resources, data curation, formal analysis, validation, investigation, methodology, project administration, and writing—original draft.

H Sabe: resources, supervision, funding acquisition, project administration, and writing—original draft, review, and editing.

## Conflict of Interest Statement

The authors declare that they have no conflict of interest.

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
