## [Reviewer comments · Life Science Alliance]

Life Science Alliance

p53 ensures the normal behavior and modification of G1/S-specific histone H3.1 in the nucleus

Tsukasa Oikawa, Junya Hasegawa, Haruka Handa, Naomi Ohnishi, Yasuhito Onodera, Ari Hashimoto, Junko Sasaki, Takehiko Sasaki, Koji Ueda, and Hisataka Sabe

DOI: <https://doi.org/10.26508/lsa.202402835>

Corresponding author(s): Tsukasa Oikawa, Hokkaido University

Review Timeline:

Submission Date:	2024-05-22
Editorial Decision:	2024-05-23
Revision Received:	2024-05-30
Editorial Decision:	2024-05-31
Revision Received:	2024-06-05
Accepted:	2024-06-06

Transaction Report:

Please note that the manuscript was previously reviewed at another journal and the reports were taken into account in the decision-making process at *Life Science Alliance*. Since the original reviews are not subject to Life Science Alliance's transparent review process policy, the reports and author response cannot be published.

May 23, 2024

Re: Life Science Alliance manuscript #LSA-2024-02835-T

Dr. Tsukasa Oikawa
Hokkaido University Graduate School of Medicine
Department of Molecular Biology
North 15, West 7, Kita-ku,
Sapporo, Hokkaido 060-8638
Japan

Dear Dr. Oikawa,

Thank you for submitting your manuscript entitled "p53 secures the normal behavior of H3.1 by regulating nuclear phosphatidic acid and EZH2 during the G1/S phase" to Life Science Alliance. We invite you to re-submit the manuscript, revised to include the data and clarifications outlined in your most recent rebuttal.

Thank you for this interesting contribution to Life Science Alliance. We are looking forward to receiving your revised manuscript.

Sincerely,

B. MANUSCRIPT ORGANIZATION AND FORMATTING:

May 31, 2024

RE: Life Science Alliance Manuscript #LSA-2024-02835-TR

Dr. Tsukasa Oikawa
Hokkaido University
Department of Molecular Biology
North15, West7
Sapporo, Hokkaido 060-8638
Japan

Dear Dr. Oikawa,

Thank you for submitting your revised manuscript entitled "p53 ensures the normal behavior and modification of G1/S-specific histone H3.1 in the nucleus". We would be happy to publish your paper in Life Science Alliance pending final revisions necessary to meet our formatting guidelines.

- please be sure that the authorship listing and order is correct
- please remove tracked changes from the manuscript file

FIGURE CHECKS:

- the green and red channels do not seem to be showing well in Figure 1F and 1I, the colors are only visible in the merged panels
- please indicate in the video legends more clearly what videos A-C show

LSA now encourages authors to provide a 30-60 second video where the study is briefly explained. We will use these videos on social media to promote the published paper and the presenting author (for examples, see <https://docs.google.com/document/d/1-UWCfbE4pGcDdcgzcmiuJI2XMBJnxKYeqRvLLrLS08s/edit?usp=sharing>). Corresponding or first-authors are welcome to submit the video. Please submit only one video per manuscript. The video can be emailed to contact@life-science-alliance.org

A. FINAL FILES:

B. MANUSCRIPT ORGANIZATION AND FORMATTING:

Sincerely,

June 6, 2024

RE: Life Science Alliance Manuscript #LSA-2024-02835-TRR

Dr. Tsukasa Oikawa
Hokkaido University
Department of Molecular Biology
North15, West7
Sapporo, Hokkaido 060-8638
Japan

Dear Dr. Oikawa,

Thank you for submitting your Research Article entitled "p53 ensures the normal behavior and modification of G1/S-specific histone H3.1 in the nucleus". It is a pleasure to let you know that your manuscript is now accepted for publication in Life Science Alliance. Congratulations on this interesting work.

DISTRIBUTION OF MATERIALS:

Again, congratulations on a very nice paper. I hope you found the review process to be constructive and are pleased with how the manuscript was handled editorially. We look forward to future exciting submissions from your lab.

Sincerely,
